# Complexation with Polysaccharides Enhances the Stability of Isolated Anthocyanins

**DOI:** 10.3390/foods12091846

**Published:** 2023-04-29

**Authors:** Wenyi Fu, Shiyu Li, Harrison Helmick, Bruce R. Hamaker, Jozef L. Kokini, Lavanya Reddivari

**Affiliations:** Department of Food Science, Purdue University, 745 Agriculture Mall Drive, West Lafayette, IN 47907, USA

**Keywords:** anthocyanins, polysaccharides, complexation, stability

## Abstract

Isolated anthocyanins have limited colonic bioavailability due to their instability as free forms. Thus, many methods have been fabricated to increase the stability of anthocyanins. Complexation, encapsulation, and co-pigmentation with other pigments, proteins, metal ions, and carbohydrates have been reported to improve the stability and bioavailability of anthocyanins. In this study, anthocyanins extracted from purple potatoes were complexed with four different polysaccharides and their mixture. The anthocyanin–polysaccharide complexes were characterized using a zeta potential analyzer, particle size analyzer, scanning electron microscopy, and Fourier-transform infrared spectroscopy. Complexes were subjected to simulated digestion for assessing the stability of anthocyanins. Furthermore, complexes were subjected to different pH conditions and incubated at high temperatures to monitor color changes. A Caco-2 cell monolayer was used to evaluate the colonic concentrations of anthocyanins. In addition, the bioactivity of complexes was assessed using LPS-treated Caco-2 cell monolayer. Results show that pectin had the best complexation capacity with anthocyanins. The surface morphology of the anthocyanin–pectin complex (APC) was changed after complexation. APC was more resistant to the simulated upper gastrointestinal digestion, and high pH and temperature conditions for a longer duration. Furthermore, APC restored the lipopolysaccharide (LPS)-induced high cell permeability compared to isolated anthocyanins. In conclusion, complexation with pectin increased the stability and colonic bioavailability and the activity of anthocyanins.

## 1. Introduction

Natural phytochemicals, such as anthocyanins, are well known for their anti-carcinogenic, antioxidant, and anti-inflammatory effects. In plants, anthocyanins are stored in cellular vacuoles [1] and mainly exist in the form of glycosides that impart red, purple, and blue colors to fruits, vegetables, and flowers [2,3]. Isolated anthocyanins are less stable than those contained in the whole food matrix and are susceptible to degradation due to changes in environmental factors, such as pH, temperature, and light, which reduces their health promoting properties. Anthocyanins are released from the food matrix in the human gastrointestinal (GI) tract as the food is digested by endogenous digestive enzymes [4]. Unabsorbed anthocyanins reach the colon and serve as substrates for gut microbiota metabolism. Moreover, the gut bacterial metabolites of anthocyanins are shown to be effective in reducing inflammation compared to parent anthocyanins [5].

Anthocyanins that are present within a whole food matrix are more stable because of their interaction with the cell wall polysaccharides [6]. For example, a simulated GI digestion study revealed that the stability of anthocyanins was significantly higher in red cabbage than in extracts [7]. For those extracted anthocyanins, various techniques, such as encapsulation, emulsification, gelation, and complexation have been designed for stability enhancement and anthocyanins delivery to a specific location within the GI system [8]. A study reported that the degradation rate of cyanidin-3-O-glucoside was decreased under simulated GI conditions after complexation with β-cyclodextrin [9]. Biopolymers, phenolic compounds, and metal ions are also used to stabilize anthocyanins [8]. Pectin extracted from blueberry powder increased the stability of anthocyanidins under GI simulation when complexed with three different anthocyanidin standards [10]. Moreover, anthocyanins from strawberries encapsulated with inulin were stable at high temperatures [11]. Many polysaccharides, especially dietary fibers, have been reported to ameliorate gut health by reducing gut bacterial dysbiosis [12], inflammation [13], and improving gut barrier function [14]. However, the complexation efficiency of different polysaccharides and potato anthocyanins, and the role of complexes in improving anthocyanin stability and bioactivity, are not known. Purple potatoes are an important source of anthocyanins and exhibited anti-inflammatory [15], anti-colitic [16], anti-hypertensive [17], and anti-diabetic effects [18] in a whole food matrix.

Our study is focused on the complexation of extracted anthocyanins from purple potatoes with different polysaccharides, including pectin, inulin, starch, cellulose, and their mixture in equal proportions. The objectives of this study were to (1) characterize the structure of polysaccharide and anthocyanin complexes and (2) identify the complex that improves the stability of anthocyanins both in terms of concentration of free and bound anthocyanins after simulated digestion and the ability to reduce colon cell permeability.

## 2. Materials and Methods

### 2.1. Materials

Purple potatoes (*Solanum tuberosum* L. var. Purple Majesty) were harvested from San Luis Valley Experiment Station (Center, CO, USA). They were washed, baked (oven; 180 °C, 50 min), freeze-dried (VirTis Ultra 35L Pilot Lyophilizer; Warminster, PA, USA), and stored at −80 °C for further use. 

### 2.2. Polyphenol Extraction and Anthocyanin Fractionation

The extraction of polyphenols from freeze-dried purple potato powder (10 g) was carried out using 25 mL 80% methanol containing formic acid (pH = 3.0), followed by vortexing for 10 min and incubation on ice. After 30 min, solutions were vortexed for another 5 min before centrifugation (10,000 rpm, 10 min, 4 °C). For bound polyphenol extraction, sample residues were subjected to acid hydrolysis using 2% formic acid [19]. The concentrated extracts were fractionated on a Hypersep C18 cartridge (2000 mg, Thermo Fisher Scientific, Waltham, MA, USA) that had already been activated by 80% methanol and equilibrated with acidified water. The elution of extracts was performed via three different solutions: (1) 0.001% HCl-acidified water for eluting sugars, (2) ethyl acetate to elute phenolic acids, and (3) 80% methanol (pH = 3.0) to elute anthocyanins. The anthocyanin fractions were collected, evaporated, and stored at −80 °C for further analyses [20]. Anthocyanin fractions were analyzed via UPLC (Appendix A). Petunidin was the major anthocyanin in the fraction, followed by malvidin. Minor concentrations of phenolic acids were observed and this result is in line with a previous study [21].

### 2.3. Determination of Total Phenolic Content 

Total phenolic content (TPC) (n = 6) was determined by using a Folin–Ciocalteu reagent (Sigma-Aldrich, St. Louis, MO, USA) [19]. The reagent (0.2 M) and Na_2_CO_3_ (7.5%) were allowed to react with samples, and the intensity of the blue color due to the product molybdenum–tungsten blue [22] was measured at 765 nm by a UV-Visible spectrophotometer with gallic acid as a standard.

### 2.4. Determination of Total Anthocyanin Content

The pH-differential method [23] was applied for the determination of monomeric anthocyanin content (MAC) (n = 6). Potassium chloride buffer (0.025 M; pH = 1.0) and 0.4 M sodium acetate (pH = 4.5) were added to samples separately under an appropriate dilution factor (DF). The absorbances were measured at 525 nm and 700 nm using a plate reader. MAC was calculated using the following formula.
$$MAC\;\left(\frac{mg}{L}\right)=\frac{A\times MW\times DF\times 1000}{\varepsilon \times 1}$$
where
$$A=Absorbance={\left(A_{525}-A_{700}\right)}_{pH1.0}-{\left(A_{525}-A_{700}\right)}_{pH4.5}$$
$$\varepsilon =Molar\;absorptivity=26,900\;\frac{L}{mol\cdot cm}$$

The results were expressed as cyanidin-3-glucoside equivalents (molecular weight; MW = 449.2). 

### 2.5. Complexation

The complexation of anthocyanins with polysaccharides (apple pectin (93854), chicory inulin (I2255), potato starch (33615), colloidal microcrystalline cellulose (435244) (from Sigma-Aldrich, St. Louis, MO, USA) and the mixture of four polysaccharides (1:1:1:1, *w*/*w*/*w*/*w*) was performed, according to the method of Li et al. [24]. Different ratios of anthocyanins to polysaccharides (1:5, 1:20, 1:50, and 1:100, *w*/*w*) were used, and it was found that the loss of anthocyanins after upper GI digestion was lowest at 1:20. Complexation was conducted by dispersing samples into HCl-acidified water (pH = 2.0) at 1:20 (anthocyanins: polysaccharide) ratio and shaken overnight at 4 °C. Complexes were freeze-dried and stored at −80 °C for further experiments.

### 2.6. Characterization of Complexes 

#### 2.6.1. Zeta Potential and Dynamic Light Scattering (DLS) Analyses

The zeta potential and DLS measurements were performed using a Zetasizer Nano ZS (Zetasizer Nano ZS, Malvern Instruments Inc., Malvern, UK). All the zeta potential (n = 3) and DLS measurements (n = 6) were carried out at 25 ℃. Zeta potential was read from pH 12.00 to 1.00 in 1.00-unit increments with a 0.20 pH unit tolerance and three readings per pH value using an MPT-2 Autotitrator (Malvern Instruments Inc., Malvern, UK). The samples (polysaccharides: 5 mg/mL; anthocyanins: 0.25 mg/mL) were suspended in water for zeta potential and DLS readings [25]. All measurements for zeta potential and DLS were conducted in triplicate and the averages were reported. 

#### 2.6.2. Scanning Electron Microscopy (SEM)

The morphological characterization of complexes obtained under optimized conditions was achieved via coating with gold/palladium using a sputter coater followed by SEM analysis (FEI Nova NanoSEM). 

#### 2.6.3. FTIR Spectroscopy

Fourier-transform infrared spectroscopy (FTIR) was specifically performed for the identification of bonds using an FTIR spectrometer (Nicolet Nexus 670 FTIR, Thermo Fisher Scientific, Waltham MA) with 34 scans at a resolution of 4 cm^−1^. 

### 2.7. Stability of Anthocyanins

#### 2.7.1. In Vitro Digestion

Simulated digestion was performed following the method described by Brodkorb et al. [26] with some modifications. This digestion model was composed of three fractions, namely the oral, gastric, and intestinal fractions, which imitate the human upper GI tract. For the oral phase, simulated salivary fluid (SSF) and salivary amylase (75 U/mL; Sigma-Aldrich) were added into the dispersions and gently shaken at 37 °C for 2 min. The samples were then adjusted to pH 2.0, added to simulated gastric fluid (SGF) with pepsin (4 mg/mL; Sigma-Aldrich) and incubated at 37 °C for 120 min in the gastric phase. The mixtures were then neutralized in the intestinal phase with simulated intestinal fluid (SIF), bile salts (10 mM; Sigma-Aldrich), and trypsin (10 mg/mL; Sigma-Aldrich). Following incubation at 37 °C for 120 min, HCl was applied for enzyme inhibition followed by dialysis (Spectra/Por^®^3 Dialysis Membrane; MWCO: 3.5 kD; Fisher Scientific) for 2 days in water. Samples that remained in the dialysis bag were freeze-dried and quantified for TPC and MAC. 

#### 2.7.2. Complexation Efficiency

The complexation efficiency was calculated using the loss of the TPC and MAC before and after upper GI digestion (n = 4). The equation was expressed as:$$TPC\;complexation\;efficiency\;\left(\%\right)=\frac{\left(TPC_{before}-TPC_{after}\right)}{TPC_{before}}\times 100$$
$$MAC\;complexation\;efficiency\;\left(\%\right)=\frac{\left(MAC_{before}-MAC_{after}\right)}{MAC_{before}}\times 100$$

#### 2.7.3. Effect of pH on the Stability of Anthocyanins

Isolated anthocyanins and APC (anthocyanin-pectin complex) were dissolved in water with different pHs adjusted by HCl (pH 2, 3, 5, and 7). All the samples were transferred into glass vials to observe the initial color at room temperature (RT). After 4 h at RT, pictures were captured to demonstrate the color change. Then, samples were heated by setting the plate heater to 95 °C for 6 h and the color differences were recorded. The temperature was lowered to 40 °C for color observation at 24 h and 96 h. 

### 2.8. Cell Culture

Human colon epithelial cells, Caco-2 (ATCC^®^HTB-37), with 10–30 passages were used. The cells were grown in Dulbecco’s Modified Eagle’s Medium (DMEM) with 4.5 g/L glucose, L-glutamine, and sodium pyruvate, supplemented with 10% (*v*/*v*) fetal bovine serum (FBS), 1% (*v*/*v*) MEM non-essential amino acids solutions (100×), and 1% (*v*/*v*) penicillin-streptomycin (10,000 units penicillin and 10 mg streptomycin/mL) under a humidified atmosphere of 5% CO_2_ at 37 °C. The Caco-2 cells were seeded onto a 12-well filter Transwell insert at a density of about 1 × 10^6^ cells/cm^2^/insert in DMEM cell culture medium (0.5 mL apical and 1.5 mL basolateral). The culture medium was changed every other day for 14 days. All chemicals were obtained from Sigma-Aldrich and Fisher Scientific.

#### 2.8.1. Anthocyanin Colonic Concentration of Complexes

On day 15, anthocyanins and complexes were added into the apical chamber. Later, media from the basolateral chambers were collected for anthocyanin quantification after 24 h. 

#### 2.8.2. Effects of Complexes in LPS-Induced High Cell Permeability

The differentiated Caco-2 cell monolayers were pre-treated with lipopolysaccharide (10 μg/mL; LPS; Sigma-Aldrich) on the basolateral for 72 h to induce damage and increase permeability. At the same time, a fluorescently labeled small molecule FITC (Fluorescein isothiocyanate)-dextran (5 mg/mL; Sigma-Aldrich) was added to the apical to monitor damage. On day 4, LPS was washed, and the cells were treated with anthocyanins (20 μg/mL), pectin (10 μg/mL), and APC (20 μg/mL anthocyanins; 10 μg/mL pectin). After 24 h, the medium was collected from the lower compartment and the FITC levels were measured to calculate the cell permeability (n = 8). 

### 2.9. Statistical Analysis

Values were expressed as mean ± SEM. Statistical analyses were performed by GraphPad 8, using one/two-way ANOVA and Tukey’s multiple comparisons. When *p* < 0.05, the difference was considered significant. 

## 3. Results

### 3.1. Characterization of Complexes

#### 3.1.1. Zeta Potential

Figure 1 shows the zeta potential of polysaccharides, anthocyanins, and complexes in aqueous solution at different pH. Anthocyanin zeta potential decreased from 6.03 ± 0.2 to −6.74 ± 0.3 mV as the pH was increased from 2.0 to 3.3. Pectin (−1.06 ± 0.3 mV), inulin (−2.71 ± 0.2 mV), starch (−1.78 ± 0.2 mV), cellulose (−0.093 ± 0.4 mV), and a mixture (−1.33 ± 0.4 mV) showed a slightly negative zeta potential at pH = 2.0. As expected, all the particles were negatively charged at a high pH, and the zeta potential of anthocyanins, polysaccharides, and complexes remained steady. Furthermore, AIC and ACC showed a sharp decrease in zeta potential at pH 3–5 and pH 3.5–5.5, respectively.

#### 3.1.2. Particle Size and Polydispersity Index through Dynamic Light Scattering

The particle size (Z-average) and polydispersity index (PDI) of samples were measured using a Zetasizer (Table 1). Isolated anthocyanins had the expected smallest particle size (296.6 ± 7 nm), which is intuitive when observing its relatively small chemical structure. Among the polysaccharides, cellulose had the largest particle size (21,080 ± 1200 nm) and inulin had the smallest particle size (685.0 ± 24 nm). After complexation, the particle size of APC and AMC were increased from 1299 ± 14 nm to 1327 ± 12 nm and from 556.2 ± 20 nm to 658.6 ± 40 nm, respectively. Increased particle size may indicate the possibility of agglomeration between anthocyanins and pectin. The PDI value of APC (0.39 ± 0.03) was lower than pectin (0.44 ± 0.002) and anthocyanins (0.45 ± 0.03). In addition, the particle size of AIC, ASC, and ACC was significantly decreased, while the PDI of AIC and ACC exhibited a significant increase. These observations might signify the failure in bond formation between anthocyanins and inulin, starch, and cellulose, but further experimental evidence is still needed. 

#### 3.1.3. Scanning Electron Microscopy (SEM)

SEM micrographs revealed a smooth spherical structure of freeze-dried anthocyanin extracts from purple potatoes (Figure 2). The particle size of anthocyanins was 296.6 nm (Table 1), which is in line with the SEM observation. The surface morphology of different polysaccharides, polysaccharides, and anthocyanin mixtures (APM, AIM, ASM, ACM, and AMM) and complexes were shown in Figure 3. Pectin (Figure 3A) had a flake and filament structure, while starch (Figure 3G) exhibited smooth spherical and oval shapes. Furthermore, cellulose (Figure 3J) and pectin appeared to contain additional features (irregular and rough surfaces). Ruffled structures (Figure 3D) were observed in inulin, which was similar to the SEM micrographs of commercial inulin powder reported by Zarroug et al. [27]. Anthocyanin particles were found on the surface of anthocyanin–polysaccharide mixtures. After complexation, APC (Figure 3C) exhibited a smooth and glossy sheet structure and AIC (Figure 3F) displayed rough sheet structure. In addition, starch granules remained unaltered (Figure 3I) and cellulose granules were aggregated (Figure 3L).

#### 3.1.4. Structural Analysis Using Fourier-Transform Infrared Spectroscopy (FTIR)

FTIR was employed to analyze the molecular structures of anthocyanins, polysaccharides, and complexes (Figure 4 and Appendix A). The FTIR spectrum of anthocyanins extracted from purple potatoes displayed characteristic bands associated with hydroxyl group (3236 cm^−1^), aromatic rings (1601 cm^−1^ (C=C), 1514 cm^−1^ (C=C), 1445 cm^−1^ (C-H), and 1332 cm^−1^ (C-H) vibrations), and glycosidic bond (1037 cm^−1^ (C-O) vibration). The FTIR spectra of pectin and APC showed the presence of a C=O ester stretching vibration (around 1730 cm^−1^) corresponding to the carboxyl group and a C-O-C bond stretching vibration (1223 cm^−1^) associated with glycosidic bonds in the galacturonic acid residues. Furthermore, the C-O-C bond stretching vibration (1070 cm^−1^), corresponding to the ester linkage between the carboxyl group and the methyl or acetyl group [28], was also detected. The two characteristic aromatic rings observed in anthocyanins were also found in the FTIR spectrum of the APC. However, no new bond stretching vibration was found in the FTIR spectrum of the APC. 

### 3.2. Stability of Anthocyanins in Complexes

#### 3.2.1. Effect of Simulated Digestions on Anthocyanin Stability in Complexes

The complexes were subjected to simulated digestion. The loss percent of phenolics and anthocyanins of complexes after digestion is presented in Figure 5. Although all the polysaccharides significantly improved the anthocyanin concentration, APC exhibited the lowest loss percentage in phenolics (16.00%) and anthocyanins (47.13%), suggesting that pectin had the best binding capacity with anthocyanins compared to other polysaccharides. Table 2 reports the content of total phenolics and anthocyanins at different digestion phases. The total phenolic content was significantly increased after pepsin digestion in the APC group. During pancreatin-bile digestion, significant degradation of anthocyanins was observed in the anthocyanin fraction group. However, APC significantly retained the anthocyanins during this digestion process. 

#### 3.2.2. Stability of Anthocyanins with Changes in pH and Temperature

The anthocyanins and APC were subjected to pH 2, pH 3, pH 5, and pH 7 and incubated at different temperatures (Figure 6). Both anthocyanins and APC showed rose-red color and the best stability at pH = 2; however, the red color faded to pink when pH increased to 3, 5, and 7. Anthocyanins at a low pH of around 3 still maintained good stability. Anthocyanins at pH 5 changed to a grey color after 4 h of incubation and then turned to yellowish-green color at 95 °C for 10 h. At a neutral pH, color changes occurred in both anthocyanins (yellowish-green color) and APC (light grey) after 96 h at 40 °C. In summary, complexation with pectin significantly improved the color stability of anthocyanins at a high pH and temperature.

#### 3.2.3. The Colonic Concentration of Anthocyanins In Vitro

The colonic permeability of anthocyanins after the simulated in vitro upper GI digestion was measured using the Caco-2 monolayer (Figure 7). After Caco-2 cells were exposed to anthocyanins and complexes for 24 h, the colonic concentration of anthocyanins available for gut bacteria from the complexes was measured. APC (37.14 ± 0.2%), AIC (34.11 ± 0.4%), ACC (35.61 ± 0.3%), and AMC (36.06 ± 0.4%) retained anthocyanins to a significantly higher level than isolated anthocyanins (31.05 ± 0.3%). ASC (25.91 ± 0.3%) had the lowest anthocyanin retention ability among other complexes.

### 3.3. Epithelial Cell Permeability In Vitro

The FITC-dextran permeability assay using Caco-2 cell model is presented in Figure 8. The differentiated cells were treated with LPS (10 μg/mL) for three days to induce damage and increase permeability in this study. After a 72 h treatment of LPS, treatment groups including anthocyanins (20 μg/mL), APC (20 μg/mL anthocyanins; 10 μg/mL pectin), and pectin (10 μg/mL) after upper GI digestion were applied in the basolateral chamber for 24 h and 48 h. Increased FITC-dextran level was observed in the positive control group, isolated anthocyanins did not reverse the LPS-induced high permeability. However, both pectin and APC treatments significantly restored LPS-induced high permeability. 

## 4. Discussion 

In this study, we complexed anthocyanins from purple potatoes with different polysaccharides. The evaluation of the magnitude of charge between anthocyanins and polysaccharides and the particle sizes, surface morphology, FTIR spectra, stability, and retention ability of complexes helps us to determine which polysaccharides exhibited the best binding capacity with anthocyanins. The results revealed that pectin showed an outstanding ability to increase the stability of anthocyanins.

The magnitude of the electrostatic repulsion or attraction between anthocyanins and polysaccharides can affect their complexation efficiency. Zeta potential assessed the surface charge of anthocyanins, polysaccharides, and complexes at different pH conditions. Under pH at 2.0, attraction occurred between anthocyanins (positive) and polysaccharides (negative). Based on Padayachee et al., ionic interactions were found between anthocyanins and pectin [29]. When one particle is positively charged and the other one is negatively charged, the electrostatic attraction can lead to agglomeration [30]. Z-average presents the measurement of the hydrodynamic diameter of the particles. A higher Z-average in APC could be an indicator of agglomeration between anthocyanins and polysaccharides. Only APC and AMC reported increased particle size after complexation. PDI is a measure of the degree of heterogeneity in the particle size distribution of samples. A lower PDI value indicates homogeneous size distribution, and the particle sizes are uniform. APC had a reduced PDI value after complexation, showing a more uniform size distribution and a closer particle size proximity in the complex, which suggests the formation of a complex between anthocyanins and pectin. The surface morphology reveals that the anthocyanin particles were dispersed in polysaccharides (Figure 3B,E,H,K,N). After complexation, the surface textures of APC (Figure 3C), AIC (Figure 3F), and AMC (Figure 3O) were changed. Soluble fibers (pectin and inulin) appeared in sheet structure and insoluble (starch and cellulose) fibers retained the same structure after complexation at −4 °C. A study reported that particles with smooth surfaces have a smaller contact area than rough surface particles, which are less susceptible to degradation reactions [31]. 

FTIR spectroscopy is a powerful characterization technique for detecting organic molecules’ functional groups, including hydroxyls, carbonyls, esters, amides, etc. [32]. The FTIR spectra of anthocyanins, pectin, and APC were analyzed, and the specific chemical bonds and functional groups were identified at different wavelengths. In Figure 4, the stretching bands of APC resembled pectin due to the high ratio of pectin in the complex. Furthermore, most of the characteristic functional bonds of pectin and anthocyanins were detected. Apple pectin primarily consists of chains of α-1,4-D-galacturonic acid with various functional groups, including carboxyl groups that can form ester linkages. The degree of methyl esterification of these linkages can affect the functional properties of pectin [33]. Additionally, two C-O-C bond stretching vibrations were observed in the FTIR spectrum of pectin, corresponding to glycosidic bonds (1223 cm^−1^) and ester linkages (1070 cm^−1^) [34]. Anthocyanins contain aromatic rings that result in unique bond stretching vibrations [35]. Anthocyanins, pectin, and APC are ascribed to the stretching vibrations of free, inter-, and intra-molecular hydroxyl groups at similar broad bands, peaking at around 3230–3360 cm^−1^ (Figure 4). Changes in the vibrational intensity of anthocyanins indicated a greater exposure of hydrogen bonds, which suggested a possible weak hydrogen binding between anthocyanins and pectin in the APC. 

Simulated digestion is a common method for studying human upper GI tract digestion to predict the stability and bioavailability of anthocyanins. Isolated anthocyanins were more susceptible to upper GI digestion, which agrees with work that found a reduction in the stability of anthocyanins from commercial black currant juice in the intestinal fluid with pancreatin [36]. In addition, Yang et al. revealed that there was a dramatic decrease in the concentration of anthocyanins from red wine in the simulated intestinal digestion [37]. Furthermore, Podsędek et al. reported that the loss of anthocyanins in anthocyanin-rich extract from red cabbage was about 87% and only 32% loss was observed in whole cabbage [7]. Therefore, complexation as a stabilizing method is necessary to help anthocyanins resist the loss in the upper GI tract to enter the colon. The concentration of anthocyanins after digestion is highly dependent on complexation efficiency. In plant cells, pectin had the highest affinity for phenolic acids and proanthocyanins [38]. Apple pectin is a soluble fiber with a high level of methyl esterification [39]. Liu et al. reported strong hydrophobic interactions between methyl groups of pectin and the dihydropyran heterocycles (C-ring) of procyanidins [40]. However, soluble blueberry pectin (high methoxyl) showed the least binding ability with anthocyanins, as reported by Lin et al. [41]. This suggests that the interactions may likely be driven through electrostatic interactions of oppositely charged particles, which was confirmed through zeta potential. 

Isolated anthocyanins were unstable, and their structural stability was altered by the changes in pH and temperature [42]. At low pH conditions, anthocyanins are electron donors and form flavylium cation, which results in them being highly soluble in water but sensitive to pH changes [43]. Previous work reported that the stability of anthocyanins at acidic conditions is higher [44]. With increasing pH, anthocyanins and APC experience color fading. At a neutral pH, anthocyanins show a purple hue and then turn to blue at an increasing pH [45]. In addition, APC showed significantly better stability than isolated anthocyanins with increased pH and temperatures. As Buchweitz et al. reported, the stability of anthocyanins was significantly improved by pectin compared to the anthocyanin extracts under 20 °C conditions for 18 weeks [46]. In conclusion, APC can stabilize anthocyanins at high pH and temperature conditions. 

Many studies used the Caco-2 cell monolayer to investigate anthocyanin absorption [47]. Cheng et al. reported the increased intestinal absorption of anthocyanins with an anthocyanin–phospholipid complex [48]. A higher anthocyanin concentration in the basolateral chamber indicates higher permeability and less complexation. Thus, the disruption of the complex was quantified by a high permeability percentage. It was expected that the isolated extracts had a lower retention, indicating a lower abundance in the colon. APC exhibited a significantly higher retention of anthocyanins compared to AIC, ASC, and ACC. Pectin might be considered a good candidate to improve the colonic concentrations of isolated anthocyanins. 

Differentiated Caco-2 cells with an LPS-induced high permeability model were used to determine the role of isolated and complexed anthocyanins in colonic cell permeability. The permeability of the gut barrier was monitored via the FITC-dextran concentration in the basolateral chamber. FITC–dextran measured in anthocyanin-treated wells matched the concentration detected in only LPS-treated wells, which explained why the isolated anthocyanins failed to restore the barrier function. The higher the FITC concentrations, the higher the permeability and barrier dysfunction [49]. Although many studies have reported that the isolated anthocyanins from plant foods inhibit the disruption of the Caco-2 intestinal barrier function [50], researchers suggested that the protective effects of anthocyanin-rich extracts in barrier integrity are positively correlated with cyanidin and delphinidin but not malvidin and peonidin [51]. Petunidin and malvidin are two major anthocyanidins in purple potatoes [16], which might explain why the anthocyanins did not restore intestinal permeability in the LPS-induced Caco-2 monolayer model. Moreover, we used the extracts after in vitro digestion, where the anthocyanins were exposed to pH changes, rendering them ineffective. The barrier function was restored by pectin and APC treatment at 24 and 48 h. The effect of APC in reducing the LPS-induced permeability is due to pectin in the complex rather than the anthocyanins.

## 5. Conclusions

In this work, we have shown that pectin exhibited the highest complexation efficiency with isolated anthocyanins from purple potatoes compared to inulin, starch, cellulose, and a mixture of these four polysaccharides. Anthocyanins complexed with pectin at pH 2.0 enhanced the stability of anthocyanins when subjected to in vitro digestion and a wide range of pH and temperature conditions. Complexation also increased the colonic concentrations of anthocyanins in vitro. Additionally, APC significantly restored LPS-induced increase in gut permeability. Overall, our results suggest that pectin is the best candidate for complexation with anthocyanins, offering improved gastrointestinal stability and demonstrating potential therapeutic effects on the restoration of high gut permeability. These findings highlight the potential use of APC as a functional food ingredient or nutraceutical for gut health.

## Figures and Tables

**Figure 1 foods-12-01846-f001:**
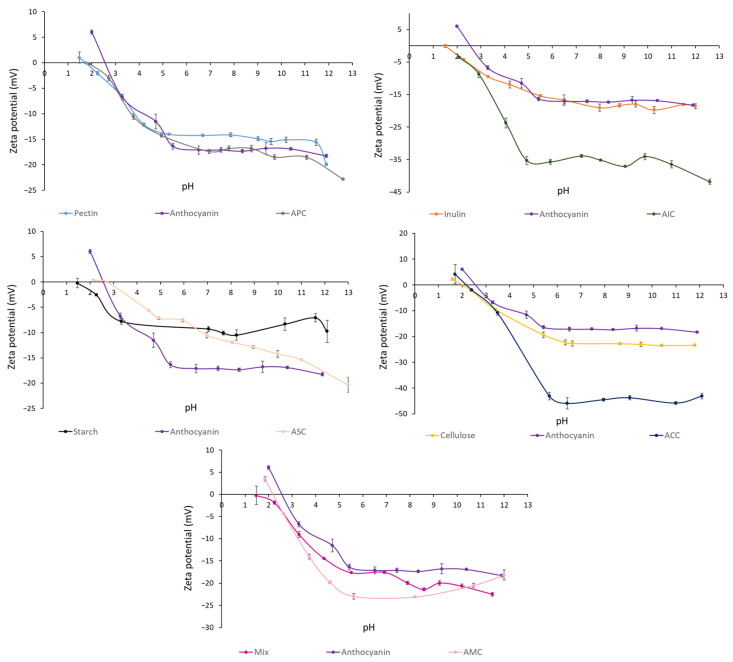
Zeta potential (mV) of polysaccharides, isolated anthocyanins from purple potato, and complexes at different pH levels. Values are shown by means ± SEM (n = 3). APC, anthocyanin–pectin complex; AIC, anthocyanin–inulin complex; ASC, anthocyanin–starch complex; ACC, anthocyanin–cellulose complex; AMC, anthocyanin mixture of polysaccharides complex.

**Figure 2 foods-12-01846-f002:**
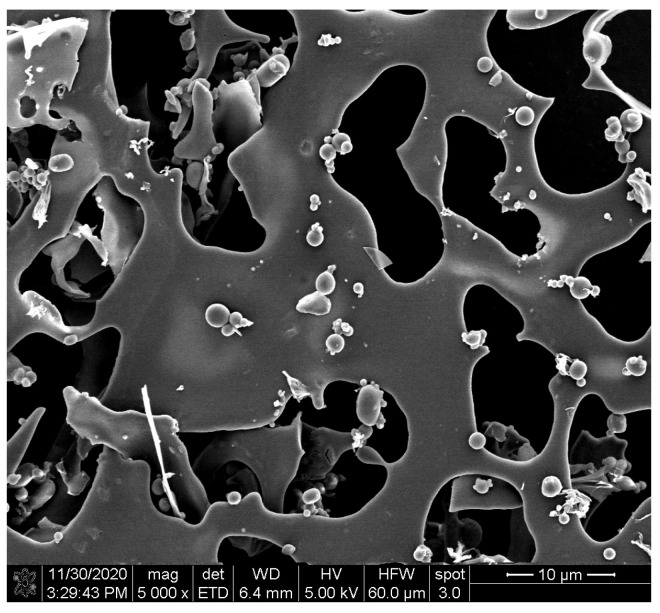
Scanning electron microphotograph of freeze-dried isolated anthocyanins from purple potatoes.

**Figure 3 foods-12-01846-f003:**
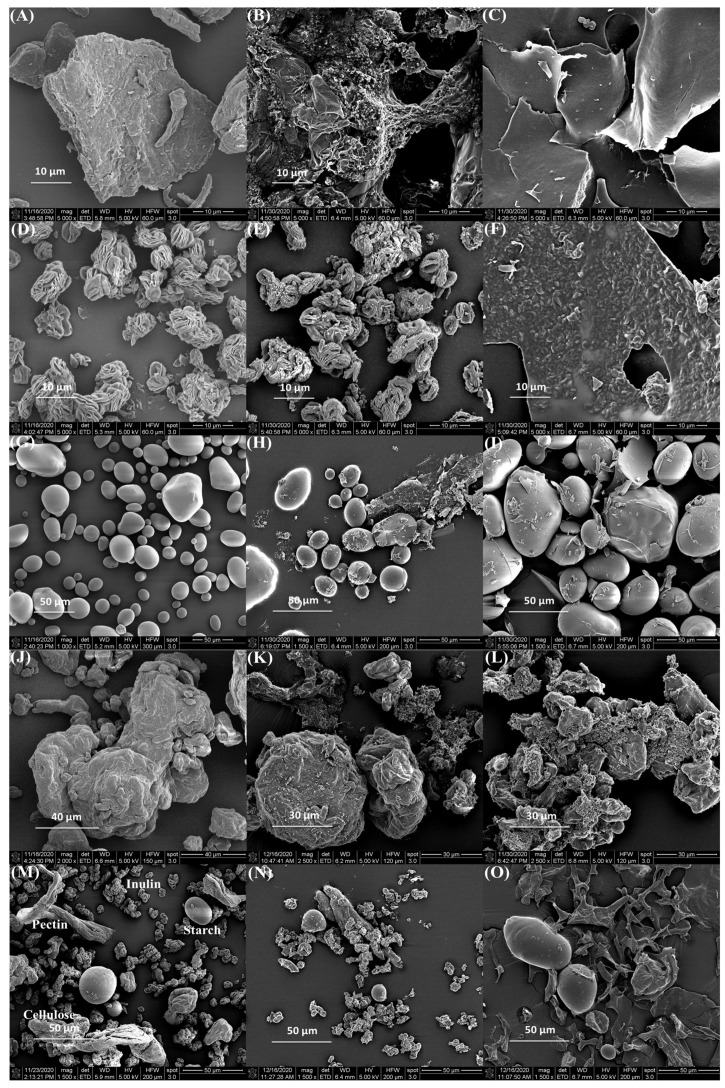
Scanning electron microphotographs of (**A**) pectin; (**B**) APM; (**C**) APC; (**D**) inulin; (**E**) AIM; (**F**) AIC; (**G**) starch; (**H**) ASM; (**I**) ASC; (**J**) cellulose; (**K**) ACM; (**L**) ACC; (**M**) mixture of four polysaccharides; (**N**) AMM; (**O**) AMC. In the triplet code, the first letter A is anthocyanin; the third letter M is a mixture; C is complex; the middle letters P, I, S, C, and M correspond to pectin, inulin, starch, cellulose, and a mix of all polysaccharides, respectively.

**Figure 4 foods-12-01846-f004:**
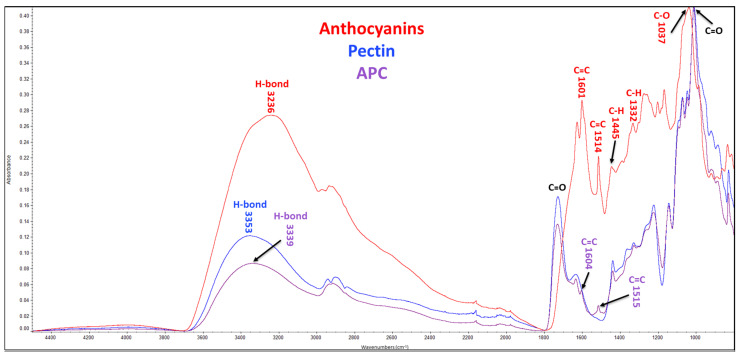
FTIR spectra. APC, anthocyanin–pectin complex.

**Figure 5 foods-12-01846-f005:**
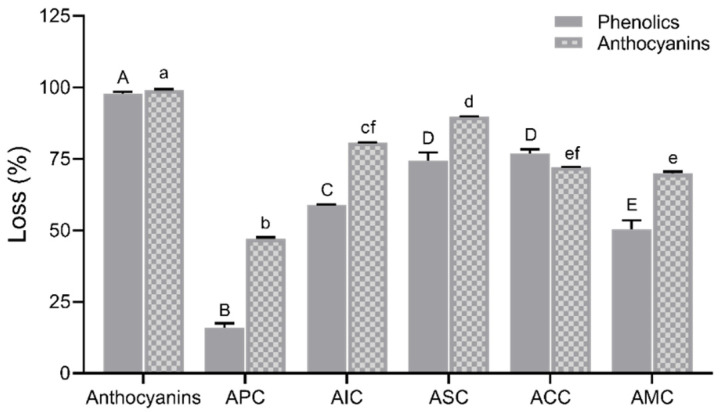
Loss in percentage of phenolics and anthocyanins after simulated digestion. Values are shown as means ± SEM (n = 4), and different letters on the bars indicate differences between polysaccharides in complexes for phenolics or anthocyanins at *p* < 0.05. APC, anthocyanin–pectin complex; AIC, anthocyanin–inulin complex; ASC, anthocyanin–starch complex; ACC, anthocyanin–cellulose complex; AMC, anthocyanin mixture of polysaccharides complex.

**Figure 6 foods-12-01846-f006:**
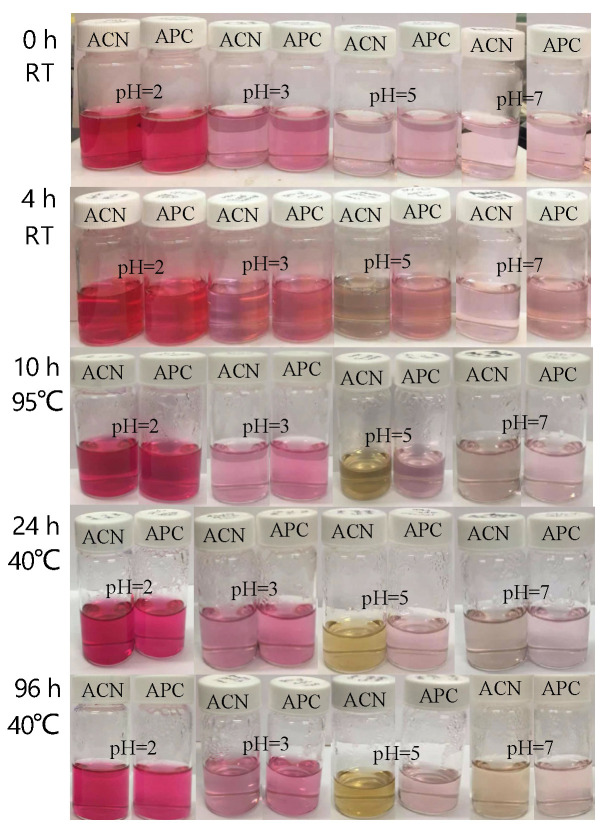
The intensity of anthocyanins in extracts and APC at different pHs and temperatures; ACN, isolated anthocyanins; APC, anthocyanin–pectin complex.

**Figure 7 foods-12-01846-f007:**
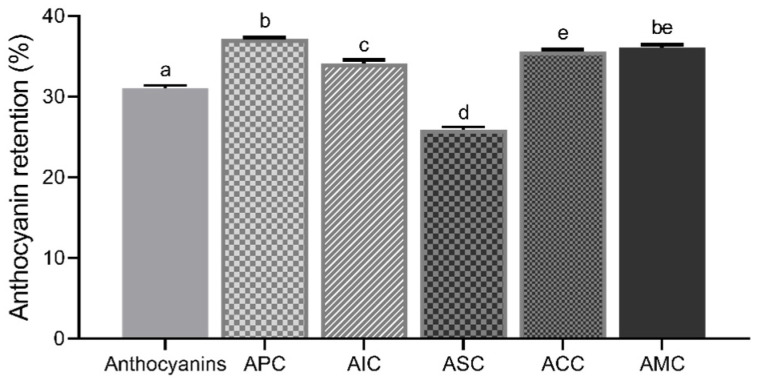
Anthocyanin retention (%) of isolated anthocyanins and complexes. Values were shown as means ± SEM (n = 8), and different letters on the bars indicate differences between the means of complexes at *p* < 0.05. APC, anthocyanin–pectin complex; AIC, anthocyanin–inulin complex; ASC, anthocyanin–starch complex; ACC, anthocyanin–cellulose complex; AMC, anthocyanin mixture of polysaccharides complex.

**Figure 8 foods-12-01846-f008:**
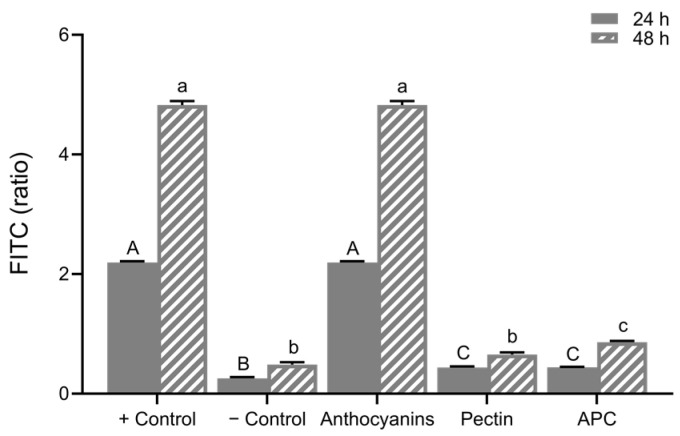
The relative FITC–dextran ratio. Caco-2 permeability at 24 h was calculated (24 h/0 h) and at 48 h (48 h/0 h). Values are shown as means ± SEM (n = 8), different letters on the bars indicate differences at *p* < 0.05. APC, anthocyanin–pectin complex.

**Table 1 foods-12-01846-t001:** Particle size and polydispersity index of polysaccharides, anthocyanins, and complexes.

Samples	Z-Average (nm)	Polydispersity Index
Pectin	1299 ± 14 ^a^	0.44 ± 0.002 ^a^
Inulin	685.0 ± 24 ^b^	0.49 ± 0.06 ^a^
Starch	724.8 ± 23 ^b^	0.50 ± 0.02 ^a^
Cellulose	21080 ± 1200 ^c^	1 ^b^
Mixture	556.2 ± 20 ^d^	0.52 ± 0.04 ^a^
Anthocyanins	296.6 ± 7 ^e^	0.45 ± 0.03 ^a^
APC	1327 ± 12 ^a^	0.39 ± 0.03 ^a^
AIC	469.2 ± 29 ^f^	0.67 ± 0.06 ^c^
ASC	434.8 ± 29 ^f^	0.50 ± 0.02 ^a^
ACC	1545 ± 119 ^a^	0.99 ± 0.005 ^b^
AMC	658.6 ± 40 ^bd^	0.44 ± 0.04 ^a^

Values are means ± SEM (n = 6). Similar letters superscript indicated differences at *p* < 0.05 using Tukey’s test. APC, anthocyanin-pectin complex; AIC, anthocyanin-inulin complex; ASC, anthocyanin-starch complex; ACC, anthocyanin-cellulose complex; AMC, anthocyanin mixture of polysaccharides complex.

**Table 2 foods-12-01846-t002:** Changes in total phenolics and anthocyanins during different digestion phases.

Digestion Phase	Total Phenolics ^1^	Anthocyanins ^2^
Anthocyanin fraction (mg/mL dry extract)		
Oral digestion	5.34 ± 0.09 ^a^	6.77 ± 0.09 ^a^
Pepsin digestion	5.29 ± 0.08 ^a^	6.66 ± 0.2 ^a^
Pancreatin-bile digestion	4.82 ± 0.07 ^b^	0.72 ± 0.02 ^b^
Pectin (mg/g)		
Oral digestion	0.05 ± 0.004 ^a^	ND
Pepsin digestion	0.12 ± 0.02 ^ab^	ND
Pancreatin-bile digestion	0.25 ± 0.01 ^b^	ND
APC (mg/mL dry complex)		
Oral digestion	2.92 ± 0.1 ^a^	1.46 ± 0.04 ^a^
Pepsin digestion	3.53 ± 0.1 ^b^	1.87 ± 0.05 ^b^
Pancreatin-bile digestion	4.81 ± 0.2 ^b^	1.33 ± 0.01 ^a^

^1^ Expressed as gallic acid equivalents (GAE); ^2^ expressed as cyanidin-3-glucoside (C3G) equivalents. Values are means ± SEM (n = 6). Superscript letters within the same column indicate differences between digestion phases for the same sample at *p* < 0.05 using Tukey’s test. APC, anthocyanin–pectin complex. ND, not detected.

## Data Availability

The data presented in this study are available on request from the corresponding author.

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
