# Peer review of "Complexation with Polysaccharides Enhances the Stability of Isolated Anthocyanins"

_foods, 2023, doi:10.3390/foods12091846_

Round 1

Reviewer 1 Report

I have read the manuscript entitled “Complexation with polysaccharides enhances the stability of isolated anthocyanins”.  The authors conducted a study of polysaccharide complexes with anthocyanins.  My main complaint is the doubtful reliability of the results obtained.

1.      The authors obtained the fraction of anthocyanins from purple potato (Solanum tuberosum). The authors do not characterize the resulting fraction. Are only anthocyanins contained in this fraction? There is no proof. For example, chromatograms should be provided and the resulting fraction should be characterized according to the results obtained. Foods journal belongs to the high-impact edition of the 1st quartile, the arguments of the authors of the publications must be irrefutable.

2.      Zeta potential. If the investigated fraction contained compounds other than anthocyanins that bind to polysaccharides, then this affected the results of determining Zeta potential. In this case, the conclusions of the authors that anthocyanins formed complexes are unreliable.

3.      Figure 1. What is an AMC complex? How was this anthocyanin mixture of polysaccharides complex obtained? There is no description of obtaining AMC complex in Materials and Methods.

4.      FTIR results are not informative. How did the authors take the spectra - in solution or in a tablet? Based on the given spectra, the fact that polysaccharides form hydrogen bonds with anthocyanins (phenols) does not follow. Figure 4 is of poor quality, the numbers are not visible. In addition, Figure 4 shows 6 patterns, and in the legend there are only five descriptions.

5.      It is not clear how Figure 5 and Table 2 are related. Do they repeat each other?

6.      Table 2. Why do the authors determine the total content of phenolic compounds, if the fraction, as they say, contains only anthocyanins?

Author Response

Comment 1. The authors obtained the fraction of anthocyanins from purple potato (Solanum tuberosum). The authors do not characterize the resulting fraction. Are only anthocyanins contained in this fraction? There is no proof. For example, chromatograms should be provided and the resulting fraction should be characterized according to the results obtained. Foods journal belongs to the high-impact edition of the 1st quartile, the arguments of the authors of the publications must be irrefutable.

Response 1: Thank you for the helpful comments. Anthocyanin fraction is enriched in anthocyanins. We added the following chromatogram to the supplementary data. The related description has been added in the manuscript (line 72-74).

Comment 2. Zeta potential. If the investigated fraction contained compounds other than anthocyanins that bind to polysaccharides, then this affected the results of determining Zeta potential. In this case, the conclusions of the authors that anthocyanins formed complexes are unreliable.

Response 2: The presence of petunidin and malvidin as the major anthocyanins in the anthocyanin fraction from purple potatoes was confirmed by UPLC. Chromatogram is presented in the supplementary data

Comment 3. Figure 1. What is an AMC complex? How was this anthocyanin mixture of polysaccharides complex obtained? There is no description of obtaining AMC complex in Materials and Methods.

Response 3: AMC refers to an anthocyanin-mixture of four polysaccharides (1:1:1:1, w/w/w/w) complex. Anthocyanins and the mixture of four polysaccharides in equal proportions were complexed according to the method of Li et al. (Materials and Methods: 2.5. Complexation).

Comment 4. FTIR results are not informative. How did the authors take the spectra - in solution or in a tablet? Based on the given spectra, the fact that polysaccharides form hydrogen bonds with anthocyanins (phenols) does not follow. Figure 4 is of poor quality, the numbers are not visible. In addition, Figure 4 shows 6 patterns, and in the legend there are only five descriptions.

Response 4: The spectra were taken by using the original forms of the polysaccharides (powder), and freeze-dried anthocyanin fraction and complexes.

We did not detect any new bond stretching vibrations in the FTIR spectrum of APC. The changes in vibrational intensity that suggest a higher exposure of hydrogen bonds, which might indicate the possibility of a weak hydrogen bonding between anthocyanins and pectin in APC. To provide a clearer picture of our findings, we reorganized the FTIR spectra of anthocyanins, pectin, and APC. The results of other polysaccharides and complexes are available in the supplementary material.

In Figure 4, the first pattern is anthocyanin, we didn’t use any abbreviations for it, so we didn’t put it in the legend.

Comment 5. It is not clear how Figure 5 and Table 2 are related. Do they repeat each other?

Response 5: They are related but targeted at different points. Figure 5 emphasizes the percent loss after all phases of in vitro digestion; Table 2 shows the quantity of anthocyanins during different phases of in vitro digestion, which look at the stability of the anthocyanin-pectin complex in the upper GI tract.

Comment 6. Table 2. Why do the authors determine the total content of phenolic compounds, if the fraction, as they say, contains only anthocyanins?

Response 6: FCR assay is used to measure the phenols, but the assay measures the reductive capacity of an antioxidant. Anthocyanins in potato are acylated and the digestion releases a lot of phenolic acids and that is why we used phenolic measurements also.

Reviewer 2 Report

The submitted article reported complexation of the variety of polysaccharides and their proportions on the stability of the anthocyanin in IG. The article was well-written and the experimental design was clearly described. However, the introduction lacked the information related to use of polysaccharide types for encapsulation of the anthocyanin. A lot of article reported the use of starch to encapsulate anthocyanins. Can author also differentiate the methods and system to give a novelty of this research?

Where is line numbers?

Purple potatoes were baked in oven 180 oC for 50 minutes. It is important to know that during the baking, the anthocyanin can degraded and lost, due the sensitivity to high temperatures. How to prevent this degradation?

What is the consideration using apple pectin, chicory inulin, potato starch, and colloidal microcrystalline cellulose? How were the polysaccharides obtained for this study? Are they extracted?

Subsection 2.5, different ratios of anthocyanins to polysaccharides (1:5, 1:20, 1:50, 1:100) are not clear. It needs to give more information related to weight of material used for the experiment. The used methods should be repeatable for other researcher.

AMC is an abbreviation for anthocyanin mixture of polysaccharides complex. It needs to elaborate ration of polysaccharides in the mixture.

Line 15 in “Discussion”, participle size?

What is FITC? Its abbreviation cannot be found in the manuscript.

In conclusion, the conclusion needs to be improved a lot. The recent conclusion did not report in the important messages obtained in the study.

Many old articles, it is suggested to use 2018-2022.

Author Response

Comment 1. Where is line numbers?

Response 1: Thank you for the comments. Line numbers are on the right of the text.

Comment 2. Purple potatoes were baked in oven 180 oC for 50 minutes. It is important to know that during the baking, the anthocyanin can degraded and lost, due the sensitivity to high temperatures. How to prevent this degradation?

Response 2: Yes, we agree that anthocyanins are sensitive to high temperature, but potatoes are almost always consumed after processing (baked, steamed, chipped, fried, boiled, or microwaved), our previous study has shown that baking is an ideal method which can largely retain the metabolites profile of color-fleshed potatoes compared to raw potatoes [1]

Comment 3. What is the consideration using apple pectin, chicory inulin, potato starch, and colloidal microcrystalline cellulose? How were the polysaccharides obtained for this study? Are they extracted?

Response 3: To capture different groups of dietary fibers and starch that are most common. They were purchased from Sigma-Aldrich, we have added the following product information in the manuscript (line 99-100).

Samples

Company

Item #

Pectin from apple                

Sigma-Aldrich corporation

93854-100G

Inulin from chicory

I2255-100G

Starch

33615-250G

Cellulose

435244-250G

Comment 4. Subsection 2.5, different ratios of anthocyanins to polysaccharides (1:5, 1:20, 1:50, 1:100) are not clear. It needs to give more information related to weight of material used for the experiment. The used methods should be repeatable for other researcher.

Response 4: Thank you for the comments. To determine the optimal ratio of anthocyanins to polysaccharides for complexation, four distinct ratios were formulated, consisting of 1 part of anthocyanins and 5, 20, 50, and 100 parts of polysaccharides (w/w). Subsequently, these complexes underwent simulated digestion and dialysis, followed by freeze-drying for the quantification of MAC.

Comment 5. AMC is an abbreviation for anthocyanin mixture of polysaccharides complex. It needs to elaborate ration of polysaccharides in the mixture.

Response 5: AMC is the abbreviation of anthocyanin-mixture of four polysaccharides (1:1:1:1, w/w/w/w) complex, we have added the related information in Method: 2.5. Complexation (line 101).

Comment 6. Line 15 in “Discussion”, participle size?

Response 6: Thanks for your comments. It should be “particle size”, we have modified the word (line 303).

Comment 7. What is FITC? Its abbreviation cannot be found in the manuscript.

Response 7: FITC is the abbreviation of Fluorescein isothiocyanate We have added the explanation of FITC in the manuscript (line 159-160), thanks for the comments.

Comment 8. In conclusion, the conclusion needs to be improved a lot. The recent conclusion did not report in the important messages obtained in the study.

Response 8: We improved the conclusions, thanks for your comments.

Comment 9. Many old articles, it is suggested to use 2018-2022.

Response 9: We have updated the old articles, thanks for your comments.

Reviewer 3 Report

This manuscript presents an original study related to the stability of an important class of bioactive compounds - anthocyanins.

The anthocyanins isolated from purple potato were complexed with several types of polysaccharides in order to increase their stability in the colon.

The authors used different techniques for complexing the extract of anthocyanins with natural polysaccharides (apple pectin, chicory inulin, potato starch), as well as with colloidal microcrystalline cellulose.

This study is very interesting and particularly useful for the development of supplements that increase the concentration of anthocyanins in the intestinal transit (especially the colon), as it is known that anthocyanins are not stable to changes in pH or heat treatment.

The results of the experiment highlighted the fact that pectin can be a good matrix for the complexation of anthocyanins, the form that can improve the concentration of anthocyanins in the colon.

The scientific quality of the manuscript it rises to the scientific level of the Foods Journal. The technical quality of the manuscript is good in terms of how it was written and how the experimental results are presented. The style of expression reflects the scientific training of the authors being in accordance with the requirements of writing the article.

The Abstract is concise and contains sufficient information to highlight the content of the article and the Introduction section provides a clear statement of the problem studied in the present manuscript.

The Materials and methods section are well presented and appropriate for the purpose of research.

Results follow the guidelines described in the Author Guidelines and they are well presented and discussed.

References are relevant and current and follow the journal’s format. However, the authors are advised to add more bibliographic references to this section because they are not enough.

The Conclusions of the article are relevant and clearly reflect the results of the study.

Author Response

Thank you so much for all your comments. We have added some more references to support our findings.

Reviewer 4 Report

foods-2287239-peer-review-v1

Complexation with polysaccharides enhances the stability of isolated anthocyanins

by Wenyi Fu et al.

The authors report on the stabilization of anthocyanins by complexation with different types of polysaccharides. The topic sounds interesting. However, some issues must be reconsidered by the authors:

Section 2.2: “extraction of polyphenols” instead of “extraction of free polyphenols”; Some more information on amounts of plant material, methanol, yield etc. would be helpful.

Section 2.3: Na2CO3 with subscript 2 and 3

Section 2.5: Headline “Determination Complexation” sounds strange; “Colloidal microcrystalline cellulose” needs some more explanation. At least the product number should be given.

Figure 1: Line thickness of the axis must be increased; they are invisible

P5: Make sure that abbreviations (AIC, ACC etc.) are properly introduced.

Table 1: Specify “Mixture”

Figure 3: The labels A), B, C).. are almost invisible. This counts for the magnification bars as well.

Figure 4: The labels are unreadable.

P8: The authors state “complexes” between the anthocyanins and the polysaccharides. Is there any conclusion that can be drawn from the FTIR measurements?

P11: cm-1 with superscript “-1”; Add some discussion about the complex formation.

References: Some references are incomplete (missing data); please check.

Author Response

Comment 1. Section 2.2: “extraction of polyphenols” instead of “extraction of free polyphenols”; Some more information on amounts of plant material, methanol, yield etc. would be helpful.

Response 1: Thank you for the helpful comments. We have added the details in the manuscript (line 65-66).

Comment 2. Section 2.3: Na2CO3 with subscript 2 and 3

Response 2: We have corrected both numbers in the manuscript (line 77), thank you for the comment.

Comment 3. Section 2.5: Headline “Determination Complexation” sounds strange; “Colloidal microcrystalline cellulose” needs some more explanation. At least the product number should be given.

Response 3: Thank you for the comments. We have corrected the headline and added the product number in the manuscript accordingly (line 99-100).

Comment 4. Figure 1: Line thickness of the axis must be increased; they are invisible

Response 4: We have added the visible axis in the manuscript (line 170), thanks for your comments.

Comment 5. P5: Make sure that abbreviations (AIC, ACC etc.) are properly introduced.

Response 5: Thanks for the comments. We had the explanations of the abbreviations you mentioned in the legend of Figure 1 (line 171-173).

Comment 6. Table 1: Specify “Mixture”

Response 6: Mixture refers to a mixture of four different polysaccharides (pectin, inulin, starch, and cellulose) in equal proportions (w/w/w/w). Specified in the methods

Comment 7. Figure 3: The labels A), B, C).. are almost invisible. This counts for the magnification bars as well.

Response 7: We have added the visible labels and magnification bars in Figure 3 (line 200), thanks for your comments.

Comment 8. Figure 4: The labels are unreadable.

Response 8: We have modified the graphs in Figure 4 (line 216).

Comment 9. P8: The authors state “complexes” between the anthocyanins and the polysaccharides. Is there any conclusion that can be drawn from the FTIR measurements?

Response 9: FTIR spectra indicates the possible bonds formed between anthocyanin and the polysaccharides in the complexes. Although we did not identify any new bond stretching vibrations in the FTIR spectrum of APC, the changes in vibrational intensity that suggest a higher exposure of hydrogen bonds, which might indicate the possibility of a weak hydrogen bonding between anthocyanins and pectin in APC. We have added the related information in the discussion (line 313-324).

Comment 10. P11: cm-1 with superscript “-1”; Add some discussion about the complex formation.

Response 10: Thanks for the comments. We have corrected the unit and added the discussion about the complex formation in the manuscript (line 301-324).

Comment 11. References: Some references are incomplete (missing data); please check.

Response 11: We have checked and corrected all the references, thanks for your comments.

[1] Blessington, T., Nzaramba, M. N., Scheuring, D. C., Hale, A. L., et al., Cooking methods and storage treatments of potato: Effects on carotenoids, antioxidant activity, and phenolics. American Journal of Potato Research 2010, 87, 479-491.

[2] Fang, H., Yin, X., He, J., Xin, S., et al., Cooking methods affected the phytochemicals and antioxidant activities of potato from different varieties. Food Chemistry: X 2022, 14, 100339.

Round 2

Reviewer 1 Report

The authors of the manuscript responded to all the comments of the reviewer, provided additional materials mentioned by the reviewer. I think that the manuscript can be accepted for publication.

Reviewer 2 Report

The article has been well-commented.